# Modelling of Stem Cells Microenvironment Using Carbon-Based Scaffold for Tissue Engineering Application—A Review

**DOI:** 10.3390/polym13234058

**Published:** 2021-11-23

**Authors:** Vieralynda Vitus, Fatimah Ibrahim, Wan Safwani Wan Kamarul Zaman

**Affiliations:** 1Department of Biomedical Engineering, Faculty of Engineering, Universiti Malaya, Kuala Lumpur 50603, Malaysia; vieralynda@gmail.com (V.V.); fatimah@um.edu.my (F.I.); 2Centre for Innovation in Medical Engineering (CIME), Department of Biomedical Engineering, Faculty of Engineering, Universiti Malaya, Kuala Lumpur 50603, Malaysia; 3Centre for Printable Electronics, Universiti Malaya, Kuala Lumpur 50603, Malaysia

**Keywords:** carbon-based, scaffold, biomaterial, biophysical factors, stem cells, tissue engineering

## Abstract

A scaffold is a crucial biological substitute designed to aid the treatment of damaged tissue caused by trauma and disease. Various scaffolds are developed with different materials, known as biomaterials, and have shown to be a potential tool to facilitate in vitro cell growth, proliferation, and differentiation. Among the materials studied, carbon materials are potential biomaterials that can be used to develop scaffolds for cell growth. Recently, many researchers have attempted to build a scaffold following the origin of the tissue cell by mimicking the pattern of their extracellular matrix (ECM). In addition, extensive studies were performed on the various parameters that could influence cell behaviour. Previous studies have shown that various factors should be considered in scaffold production, including the porosity, pore size, topography, mechanical properties, wettability, and electroconductivity, which are essential in facilitating cellular response on the scaffold. These interferential factors will help determine the appropriate architecture of the carbon-based scaffold, influencing stem cell (SC) response. Hence, this paper reviews the potential of carbon as a biomaterial for scaffold development. This paper also discusses several crucial factors that can influence the feasibility of the carbon-based scaffold architecture in supporting the efficacy and viability of SCs.

## 1. Introduction

Scaffolds are developed from a range of biomaterials and serve as platforms for cellular interaction study to stimulate the behaviours of cells. The fabrication of scaffolds is essential as it serves as a template that provides structural support for cell growth, tissue formation, and organ regeneration. In addition, it also provides biophysical and biochemical cues that mimic the natural microenvironment of cells that will influence the development of cells [1,2]. Presently, scaffolds provide a biological substitute to facilitate the repair, replacement, and regeneration of damaged tissue. Scaffolds are also used for restoring, maintaining, or enhancing the functionality of tissue damaged by trauma or diseases [3].

During the development of scaffolds, several aspects concerning the design and fabrication of the optimal scaffold architecture for tissue applications should be appropriately addressed, including; (i) the topographical architecture and mechanical properties of a bioscaffold that mimic the natural microenvironments of tissues in the body; (ii) suitable porosity and pore size that allows the flow of oxygen, nutrients, and the exchange of waste; and (iii) material biocompatibility [4]. However, biocompatibility is the most crucial factor that needs to be considered when developing scaffolds for cell studies. Materials without biocompatibility will not be favoured by cells; therefore, cells will be unable to attach and grow on them. Meanwhile, the use of non-biocompatible materials that possess some toxicity are harmful to cells and may lead to cell death or apoptosis [5]. Thus, it is essential to prioritize the biocompatibility characteristics of materials before looking into other aspects that could influence cell behaviour. The biocompatibility of scaffolds is first influenced by the types of materials used. Recently, various materials were investigated, including metal, ceramic, natural, and synthetic [6,7,8,9,10].

Among the promising biomaterials, carbon has shown excellent biocompatibility with SCs. Moreover, carbon possessed electrical properties [11], mechanical properties [12], thermal properties [13,14], corrosive resistivity [15], and excellent gas permeability [16,17]. The use of carbon in the culture of SCs is advancing, and the material has been utilised in many studies involving SC application, including human induced pluripotent stem cells (hiPSCs) [18], adipose-derived mesenchymal stem cells (AMSCs) [19], mesenchymal stem cells (MSCs) [20], cancer stem cell [21], human umbilical cord mesenchymal stem cells (HUC-MSCs) [22], neural stem cells (NSCs) [23], embryonic stem cells (ESCs) [24], and bone marrow-derived mesenchymal stem cells (BMSCs) [25]. In addition to SCs, other biological cells are also influenced by carbon materials. A recent study showed that the addition of chitosan and carbon nanotubes on polyurethane-based composite nanofibrous scaffold improved cell adhesion, the proliferation of cardiac rat myoblast cells (H9C2), and human umbilical vein endothelial cells (HUVECs) [26].

In recent years, extensive research of carbon materials has opened the opportunities to discover the potential, and various forms, of carbon. The discovery of the carbon nanostructure has attracted the interest of many researchers and it continues to grow. Carbon nano-onions (CNOs) are one of the latest class of nanomaterials, providing unique physicochemical properties which are beneficial in numerous applications [27]. Following the discovery of other classes of carbon nanostructure, the evolvement of carbon nanomaterials has been incessant. Although carbon materials have shown significant influence on various cells/tissue developments, their cytotoxicity potential remains a challenge. In regard to this, the cytotoxicity effects of carbon materials are dose dependent [28,29,30]. Thus, appropriate the concentration of carbon materials used will reduce the cytotoxicity effects while retaining a positive influence on the development of cells.

Interestingly, the different structural composition of carbon also influences the development of cells. A previous study reported that the reduced graphene oxide nanoribbons grids (rGONRs) promote slightly better MSCs proliferation compared to graphene oxide sheets and PDMS. Meanwhile, in the presence of a chemical inducer, the rGONRs enhanced the osteogenic differentiation of MSCs [31]. Another study has reported the effects of different graphene nanostructures: graphene nano-onions (GNOs), graphene oxide nanoribbons (GONRs), and graphene oxide nanoplatelets (GONPs) on AMSCs and BMSCs. Plus, only GNOs and GONPs were internalized by AMSCs while GONRs were not. Moreover, no significant effects of the different nanostructures were seen to influence the adipogenic differentiation of SCs; however, GONRs promote higher alkaline phosphatase activity compared to GNOs and GONPs [30]. Other than the structural influence of carbon, there are other crucial factors of scaffolds that play a key role in supporting and promoting cell/tissue developments. These factors, which could help modulate the SCs microenvironment, are discussed in this paper. In addition, these factors have been pointed out by various researchers as important parameters for the development of cells. However, there is still an abundance of questions that need to be addressed to determine the appropriate parameters of scaffolds for specific SC applications. Moreover, although there is much data on the different ranges of parameters, such as the size and amount of pore and stiffness that could influence specific SCs lineage differentiation, no definite or standard range has been established concerning the specifics and type of SCs. Therefore, the investigation of each parameter, in detail, is required to influence the growth, proliferation, and differentiation of SCs, and facilitate the establishment of a standard range of sizes for every parameter as a future reference for scaffold development. Thus, this paper reviews and discusses the current applications of the carbon-based scaffold in SC research and focuses on factors that support SC growth and survival.

## 2. Carbon as Scaffold Biomaterials for SC Applications

Carbon exists naturally as allotropes with distinct physicochemical properties. It is among one of the most abundant elements in the universe and is widely distributed in nature. Additionally, the human body is composed of carbon elements, where carbon is the second most abundant element after oxygen. Thus, carbon materials have become a more desirable choice in the applications of various research areas, including microscience and nanoscience, engineering, technology, material sciences, and even biomedical applications [9,32]. Current carbon materials include graphene [33], graphite [34], carbon nanotube (CNT) [20], diamond [35], fullerene [36], amorphous carbon [37], and glassy carbon [38]. Various carbon allotropes have been investigated concerning their viability as a scaffold for SCs, other biological cell applications are shown in Table 1.

Carbon, as a scaffold material for SC application, has shown promising results. However, the fabrication of a suitable external ECM as a scaffold for the cells to grow and differentiate, is still a significant challenge. The optimal architecture and properties of the scaffold, that can enhance SC survivability and differentiation into the desired functional cells, is still lacking [66]. Hence, besides biocompatibility factors, the ideal parameters for creating the scaffold, that mimic the natural ECM of SCs, are also important. Scaffolds mimicking the natural tissue ECM for SC culture may provide a more closely resembling microenvironment, similar to that of the human body, providing a better understanding of cellular response and development [67]. Therefore, the fabrication of the in vitro cell microenvironment, as in the living organism, will give a more predictive in vivo system. Hence, suitable materials, designs, and fabrication methods of microstructures for specific SCs applications, has become essential. A schematic illustration of carbon material application is presented in Figure 1.

Previously, the in vivo study played a significant role in retrieving reliable information of SC studies as the phenomenon occurs in its most natural SCs niche, inside living organisms. However, in vivo study is still insufficient. Thus, many researchers proposed 3D in vitro scaffolds, a development that closely resemble the cell microenvironment niche [68,69,70,71]. A 3D scaffold structure provides a higher surface area for cell growth and migration when compared to a 2D scaffold. Therefore, a 3D scaffold structure can be designed to achieve the most suitable physiological and structural stability. A report has shown that the architecture of the 3D carbon-based scaffold possesses a strong influence on the cells’ behaviour [72]. Undeniably, the advantages of 3D cultures provide more realistic cellular behaviour and gene expression when compared to 2D cultures [68]. Thus, the integration of carbon in SCs studies has opened significant fields in the present and future of SC studies.

Additionally, the discovery of new classes of carbon materials has opened the opportunities to discover the potential of these materials in tissue engineering applications. CNOs, also known as onion-like carbon, is part of a new emerging class of carbon that has gained popularity among researchers due to its unique properties. A study has investigated the effects of poly 4-mercaptophenyl methacrylate-carbon nano-onions conjugated with doxorubin (DOX) and loaded onto BSA nanofibers by Forcespinning methods on human fibroblast cells. This study reported that the nanofibers improved cell attachment and proliferation. Plus, the nanofiber provided a high loading capacity and could sustain a long-term release of DOX. This indicates the potential of the nanofibers as a biocompatible drug delivery system [57]. Another study group investigated the biocompatibility of an electrospun scaffold incorporated with oxidized CNOs in vivo on the subdermal of Wistar rats for 90 days. This study reported that 30 days after implantation, lymphocyte infiltration was seen but reduced in number after 90 days. However, significant inflammatory responses were observed on sample F3 (CS/PVA/ox-CNO 3.5:95.5:1) and F4 (CS/PVA/ox-CNO 5.4:92.6:2) which was believed to be due to the crystallinity of scaffold that affected the rate of enzymatic hydrolysis [73]. Meanwhile, in another study it was reported that CNOs conjugated with -glycopeptide (Gly-CNOs) and -bovine serum albumin (BSA-CNOs) does not affect the viability of NIH3T3 cells and MFC7 cells. Also, both Gly- and BSA-conjugated CNOs can be readily internalised compared to unconjugated Gly and BSA. This indicates the ability of CNOs as non-toxicity carriers [74]. Although, CNOs are remarkable in biomedical applications, investigations of their potential on SCs remain scarce. Thus, further study of CNOs on SCs is requisite. Additionally, a recent study has demonstrated an electrochemical synthesis on CNOs, reporting the capability of such a method in synthesizing hollow CNOs and solid CNOs [75]. This staggering finding should be further applied in tissue engineering. This investigation on the effects of different classes of CNOs on cells or tissue will be an exciting development.

Interestingly, carbon dots (CDs) have gained much attention from researchers due to their photoluminescent properties. CDs, also known as carbon quantum dots, graphene quantum dots, and carbon nanodots, are a zero-dimensional carbon with sizes ranging from less than 20 nm [76]. Compared to CNOs, investigations concerning the potential of CDs on stem cells and other biological cells are strongly ongoing. Shao et al. developed citric acid-based CDs for the labelling and tracking of rat BMSCs. They reported that the CDs can be internalized by BMSCs without affecting the viability of the cells, provided that the concentration of CDs is lower than 50 µg/mL. This also promotes the osteogenic differentiation of BMSCs [60]. Another study simultaneously used fish scales and hydroxyapatite nanoparticles to produce CDs by using a hydrothermal method. They reported that the resulting CDs were biocompatible with human MSCs. This suggests that they can work as fluorescent probes for bio-imaging, as they could readily enter the cytoplasm and nucleus through the cell membrane [77]. More examples of CDs are listed in Table 1. Other than the excellent photoluminescent, non-toxic, and biocompatibility properties, CDs also possess a wide range of antimicrobial activity [78]. Additionally, a study on date pit-derived carbon dots has shown to possess anticancerous effect by inhibiting the growth of human lung cancer, breast cancer (MCF-7), and PC3 cells [79].

To date, the investigation of carbon materials is blooming. Significant findings of other types of materials make it an excellent choice of material for modulating the microenvironment of SCs. Plus, it is undeniable that carbon, as a material for scaffold development, either stand-alone or mixed with other materials, possesses a significant positive effect on SCs and other biological cells. Through the availability and ability of carbon materials, an optimal SC culture system development is possible. The integration of carbon-based materials from carbon precursors in MEMS has paved a better way to fabricate 2D and 3D micro-scale carbon structures.

## 3. Carbon Precursors as Scaffold Biomaterials for SCs Applications

The use of carbon precursors as an alternative to carbon has shown more advantages and has received much attention in the production of carbon-based microstructures. In the last few years, the fabrication of carbon precursors, in various studies due to their manufacturing flexibility and customizable properties, has been employed [9]. Carbon precursor materials can be converted into high-percentage carbon materials when subjected to high temperatures or chemicals. The tuneability properties of carbon precursors have allowed the production of different patterns (i.e., organised or random alignment) that mimic the microenvironment niche of cells by using a simple method with good reproducibility and low cost. Plus, carbon precursors can be tailored to produce conductive carbon-based scaffolds through modifications to their chemical composition and pyrolysis process (Table 2). Every scaffold should exhibit mechanically and biologically suitable qualities, mimicking the ECM of SCs to support the adhesion and development of cells, something which carbon precursors can provide [80,81].

Importantly, polymers are the most common materials investigated as carbon precursors for biological application. Their unique polymer features allow the fabrication of microscale and nanoscale scaffolds with customisable pattern alignment similar to the ECM of cells [88,89]. The negative photoresist SU-8 polymer has been extensively used to produce carbon-based scaffolds for biological application and have shown to be a promising material in tissue engineering. For instance, SU-8 derived 3D carbon-based scaffolds for neural SC culture, can promote the growth of cells, and magnetic resonance assisted imaging could locate the 3D locations of the cell clusters [70]. Moreover, the SU-8-derived carbon pattern and PAN-derived carbon nanofiber can promote the adhesion and differentiation of hiPSCs -derived NSCs into neurons [90]. Plus, the polymer-derived carbon-based scaffold (i.e., resorcinol-formaldehyde (RF) gel and SU-8) could influence neural cell differentiation and proliferation [88]. However, most SCs applied on the carbon-based scaffold of polymer precursors focus on neural differentiation. Meanwhile, the knowledge of another types of SC application is still lacking or limited. Therefore, further investigations regarding the potential of different SCs, to differentiate from the lineage other cells, such as cardiac or skeletal, are required.

Other than the application of SCs on the carbon-based scaffold, other biological cells also have been investigated and provided promising results. For example, Islam et al. developed carbon micro-lattices using epoxy resin, from which it was reported that the osteoblast-like murine MC3E3-E1 cells could adhere and grow on the surface of carbon micro-lattices. Furthermore, they also hypothesised that the hollow bubble-like feature in carbon micro-lattices may direct the formation of the vasculature, one of the challenges in the biofabrication field [87]. Furthermore, non-polymer-based carbon scaffolds, such as 3D glassy carbon-based scaffolds developed from sucrose and ammonium chloride, allow the adhesion of human neuroblastoma cell lines (SH-SY5Y) and human embryonic kidney cells (HEK-293) on its surface. These scaffolds promote cell growth without using ECM ligands on its surface [72], thus indicating that the carbon-based scaffold could support the development of SCs and other biological cells.

Although carbon precursors have shown great potential as biomaterials in 3D microstructure and nanostructure fabrication, many criteria still require consideration in order to produce the optimal microstructure architecture that can provide the best stimulation towards the desired SC growth and differentiation. Hence, the scaffold design for SC application is crucial to ensure that an ideal system is achievable for future SC therapy and tissue engineering.

## 4. Factors Influencing the Behaviours of SCs

The microenvironment niche should involve the architecture, composition, signalling, and biomechanics of the cellular microenvironment, which will interact with each other to provide the necessary cues to regulate the development and functionality of cultured stem cells. Therefore, various factors should be considered to achieve a thriving stem cell culture on the carbon-based scaffold material (Figure 2). Furthermore, these factors may also influence the biocompatibility and the phenotype of stem cells towards the created scaffolds. Thus, several factors that may influence, and/or enhance, the biocompatibility of carbon-based scaffolds in stem cell culture are reviewed as follows.

### 4.1. Topology Architecture of the Scaffold

The architecture of the scaffolds affects the binding of cells through the adhesion and development of the cultured cells (Table 3). Both microscale and nanoscale architecture provide different features that could influence the cells attachment and arrangements on the scaffold (Figure 3) [90]. Therefore, the topology features of a scaffold are crucial as they help improve the cell-scaffold interaction and cells adhesion [91]. Other than that, the density of the microfabricated structure also influences the development of cultured cells. For instance, the low packing density of electrospun nanofibres significantly improved cell viability, proliferation, and infiltration compared to a tightly packed scaffold [92].

Meanwhile, Ren et al. reported that the super-aligned CNTs sheet provide better electroconductivity properties of scaffolds and promote the alignment of cardiomyocytes with elongated cell morphology, similar to the features of cardiac cells. In addition, it provides electrophysiological homogeneity on synthetic cardiac tissue [93]. Overall, the topology can enhance the protein absorption of SCs on a scaffold [52]. Therefore, it is essential to build a scaffold that could provide contact guidance for attachment, and alignment cues through its architecture, providing space for nutrients and waste product exchange, as well as allowing the infiltration of cells.

The findings in the various studies highlight the influence of surface geometry and chemistry on different scaffold materials based on the topology and surface treatment for SC culture. Hence, in short, the topology also plays a vital role in directing the development of SCs.

### 4.2. Surface Wettability of the Scaffold

Surface wettability is another crucial parameter in scaffold development, specifically for SC application. Generally, the wettability of a scaffold surface represents its hydrophobic (contact angle value of more than 90°) and hydrophilic (contact angle value of less than 90°) properties. Usually, the hydrophobic surface is not favoured by cells, while a hydrophilic surface is favourable by cells. In SC application, scaffold wettability will affect the biological response of SCs through adhesion, attachment, and proliferation on the scaffold, which will affect the biocompatibility.

Commonly, surface wettability is measured by the contact angle, which indicates the degree of water droplet interaction on the surface of the scaffold. Currently, surface roughness and the chemical composition of the scaffold are considered essential factors that modulate the wetting ability [43,99,100,101,102,103]. On the other hand, surface wettability can be improved through surface modification to increase its biocompatible activity and reduce its hydrophobic properties. For instance, Amato et al. used a reduced contact angle, indicating improved hydrophilicity after oxygen plasma treatments (Figure 4) [81].

Whether in vitro or in vivo, the cells and tissue will first contact the scaffold’s surface. Therefore, the adhesion abilities of a scaffold with cells are essential and will hugely affect the cells attachment, proliferation, and differentiation [55]. However, different scaffold materials possess different adhesion properties, and some have poor adhesion properties which are not favourable by cells. Thus, the modification should be done on the scaffold’s surface to improve its wettability and adhesion properties. Furthermore, through surface functionalisation, the hydrophilicity of scaffolds can be improved, which could enhance the favourable cellular response (Table 4).

### 4.3. Porosity and Pore Size of the Scaffold

Scaffold porosity and pore size are crucial in influencing cell migration and spreading. These factors are also important in the nutrient and waste exchange between scaffolds and the surrounding environment [92]. The pore size can be divided into three ranges: (1) nanoscale with a pore size less than 100 nm; (2) micro-scale with the pore size ranging from 100 nm to 100 µm; and (3) macroscale with a size of more than 100 µm. A pore size smaller than the cellular diameter may limit cell migration within the scaffold structure. Meanwhile, a pore size bigger than the cellular diameter allows excellent cell migration within the scaffold structure. However, cell attachment will be affected if the pore size is too large, thus negatively affecting cell spreading and differentiation. Tadyszak et al. reported that the largest pore size of carbon scaffold, at 193 m^2^/g, causes the largest decrease of cell viability compared to the other pore size (180 and 31 m^2^/g). They deduced that large pore sizes lead to increased protein absorption into the carbon pores, resulting in a higher depletion rate of nutrients from the cell media. In addition, the attachment sites were reduced with an increased pore size, as cells tend to lose their grip and escape from the scaffold structure [72].

Furthermore, several articles have also reported that porosity is vital in cell infiltration, migration, and differentiation. The scaffold’s porosity may also provide a better flow, the exchange of waste and nutrients, and the infiltration and migration of cells throughout the scaffold volume [91]. Lee et al. reported that the increase of porosity in microfibrous scaffolds increased the infiltration in neural crest cell-like synovial stem cells (NCCL-SSCs), both in vitro and in vivo, without affecting the morphology and proliferation of NCCLSSCs. The authors also found that the increased scaffold porosity level enhanced the expression of chondrogenic and osteogenic genes but subdued the expression of adipogenic and smooth muscle genes [107]. Additionally, the pore diameter for cell migration depends on the type of cells used on the scaffold. Different types of cells with different cell diameters need different pore sizes to fit in. For instance, human embryonic cell migration was blocked when the pore diameter was less than 1 µm [108]. Meanwhile, Madrid et al. found that the optimal pore size for specific bone tissue engineering ranged from 200–900 µm [109]. Table 5 shows the effect of the different pore sizes on the scaffold towards SC response. Different cells have different sizes and characteristics; therefore, the requirement for pore size and porosity level on scaffolds should be considered in the design of scaffolds. Hence, the porosity and pore size rates should be considered during microfabrication based on the types of SCs used and the direction of cell lineage differentiation for better cell growth, proliferation, and differentiation.

### 4.4. Mechanical Properties of the Scaffold

The mechanical properties of a scaffold for SC culture should have sufficient strength to function and uphold the environment according to the utilisation purpose. Fragile carbon-based scaffolds could easily break into carbon flakes or fragments, leading to toxic effects on cells [72]. Therefore, the stress resistance of the scaffold should be strong enough to prevent fracturing or breaking and to support the structural components of various applications, such as implants [119].

Mechanical properties may also influence the characteristics of the cultured SCs. The cultured SCs can mimic the stiffness of the substrates under specific environments, such as neurogenic and osteogenic environments, by developing phenotypes similar to the tissue niche stiffness [120]. SCs tend to express higher smooth muscle cell markers on soft substrates, whereas, in stiff substrates, the cells tend to express more chondrogenic and adipogenic markers. However, cells on soft substrates result in weak cell proliferation rates and spread and exhibited fewer stress fibres than the cell culture on stiff substrates [121]. Wang et al. reported that the PCL/Graphene scaffold with 48.81 kPa promoted the highest ALP activity on human AMSCs, indicating improved osteogenic differentiation compared to other scaffolds with lower elastic modulus [43].

Meanwhile, Ma et al. reported that the stiff scaffold (64 kPa) promoted better NSC attachment and proliferation than the soft scaffold (30 kPa). However, the stiffer scaffold promoted NSC adhesion, growth, and differentiation into astrocytes compared to the soft scaffold [45]. In comparison, the stiffer surface of the MWCNT/PLLA scaffold enhanced BMSC growth and differentiation [122]. In another study, Ignat et al. developed a cellulose acetate/carbon nanotubes/graphene oxide (CA-CNT-GO) scaffold. They reported that the scaffold could promote both adipogenic and osteogenic development, indicating the ability of the scaffold to support soft and hard tissue engineering [123].

Recently, a study reported that the addition of carbon nanofibers (CNFs) in alginate and gelatin hydrogel improved the Young’s moduli (534.7 ± 2.7 kPa) and electrical conductivity (4.1 × 10^−4^ ± 2 × 10^−5^ S/cm) properties; the incorporation of CNFs made it 3D printable. In addition, the resulting scaffold enhanced cellular proliferation compared to controls [124]. Meanwhile, Islam et al. developed 3D carbon architectures comprising of carbon fiber and microlattice hybrid architectures by the carbonisation of stereolithographically 3D printed epoxy microlattice architectures. These architectures were pre-filled with cotton fibre within the empty space of the microlattice structure (Figure 5). The resulting scaffold had an improved compressive strength of 156.9 ± 25.6 kPa compared to unfilled carbon microlattice architecture. The scaffolds could also promote cell proliferation within the 3D hybrid structures [125]. In comparison, Stocco et al. reported that the addition of 0.05% and 0.1% of CNT improved the Young’s modulus of electrospun nanofibre from 5.05 ± 0.03 MPa to 12.17 ± 0.32 MPa and 13.33 ± 0.41 MPa. They also reported that the 0.1% PCL/CNT had the highest stress property failure (5.65 ± 0.40 MPa) compared to 0.05% PCL/CNT (3.29 ± 0.20 MPa) and PCL (1.35 ± 0.03 MPa) [126]. Similarly, Mombini et al. reported that the tensile strength of polyvinyl alcohol/chitosan scaffolds correspond to the increased concentration of CNT added. However, an increased concentration of CNT reduced the elastic modulus of the scaffold [127].

Interestingly, a study reported that the composite material of poly-4-hydroxyphenyl methacrylate single-walled carbon nanotube/polyethylene nanocomposite (NSCT) fibre membrane shows the highest record of tensile strength (13.7 ± 3.2 GPa), Young’s modulus (243.3 ± 5.2 GPa), and toughness (1421 J g⁻^1^) compared to other reported nanocomposite; Therefore, representing the strongest and stiffest composite materials ever recorded. Moreover, the NSCT fibre supported the growth of human osteoblast cells [128]. In another study, a whey-derived porous carbon scaffold represented a porosity between 48% and 58% with diameters of pore between 1 to 400 µm. This study also reported that the compressive strength and Young’s modulus were better compared to the traditional hydroxyapatite, or tricalcium phosphate, scaffold with similar porosities. In addition, the carbon scaffold supported the viability of bone osteosarcoma cells [129]. Figure 6 shows the effect of carbon on mechanical properties of scaffold.

Additionally, a report mentioned that muscle SCs could retain their self-renew ability on soft substrates. They also possess the same characteristics as the muscle niche elasticity, which makes substrate elasticity an essential regulator in directing the fate of cultured muscle SCs [132]. Furthermore, SCs may also exhibit mechanical memory from their past physical environments with the Yes-associated protein (YAP) and transcriptional coactivator with PDZ binding domain (TAZ) as an acting intracellular mechanical rheostat that influences and controlls the fate of cell differentiation [133]. Hence, in a nutshell, the mechanical properties in the microenvironmental niche of the scaffolds are vital in regulating cell phenotypes and influencing cell differentiation.

### 4.5. Electrical Stimulation and Electroconductivity of the Scaffold

Electrotaxis is the method of applying a direct current towards cells to induce cell migration. An external electrical current could influence SC behaviour (Table 6). Under a direct electric current, SCs will move towards an anode or a cathode, whereas without an electric stimulus, SCs move randomly. Under electrical stimulation, the migration direction of SCs and migration speed is stimulated. The migration of cells through electrical stimulation is passage dependent, wherein in the late passage (p7–p10), the migration speed of the cells decreased compared to in early passage (p1–p3) cells [134]. Similarly, Hong et al. reported that at an early passage number, higher percentages of SCs were migrating to the anode under direct electric current stimulation than the percentage of cells migrating to the anode when the cells became senescent with increasing passages (Figure 7) [135].

Through electrical stimuli (ES), cell differentiation is also possible. For example, in vitro ES of NPCs activates the cells differentiation ability towards neurons [136]. Similarly, the ES could also direct the differentiation of NSCs and guide the growth of neurites [137]. Likewise, under 100 ms voltage pulses in an interval of 10 s, ES enhances neural cell proliferation and differentiation rates were obtained (Figure 8) [138]. Meanwhile, the electrical pulses could also increase the differentiation of cardiomyocytes from cardiovascular disease-specific iPSCs because the electrical cues mimic the ES in the heart [139]. Hence, ES can be used as the cues to control SC behaviour in artificial niche microenvironments towards cell differentiation and cell fate.

Additionally, electroconductive scaffolds help to support electroactive cells (i.e., cardiac cells, muscle cells, and neural cells). By considering this property, carbon is a well-known electroconductive material. Carbon materials are also electrochemically and mechanically stable in an ionic solution, which can uphold the electrical current of external electrical charge [147,148,149,150]. Furthermore, the conductivity of the carbon-based scaffold improved the communication between cells by acting as a medium to send the electrical signal from one cell to another. In other words, the electroactive cells are electrically excitable. Thus, the electroactive materials will allow the local delivery of an electrical stimulus, which influences the behaviour of SCs through proliferation, differentiation, and tissue regeneration [43,91]. For instance, the 3D environment and conductive properties of carbon-based scaffold promoted neurite elongation and better differentiation of human NSCs into dopamine-producing neurons compared to 2D electrodes [81].

Furthermore, the electroconductive scaffold also supports the growth of cardiac cells’ and has been shown to enhance the cardiogenic properties without external electrical stimulation [151]. Similarly, without ES, the conductive PCL/CNT scaffold promotes morphology elongation with the enhanced α-MYH and F-actin colocalisation of human MSCs. However, under ES, human MSCs cultured on PCL/CNT scaffold ES downregulated the α-MYH expression and inhibited their cardiomyocyte-like elongated morphology [152]. Therefore, ES may either facilitate cell differentiation or divert into the downregulation of cell marker expression, which contradicts other findings. Hence, a deeper investigation into the effect of ES on cell expression is required for a better understanding. Overall, carbon materials offer good opportunities in the fabrication of scaffolds with excellent electrical conductivity for SC research that requires conductive materials to stimulate the behaviour of SCs, either in the presence or absence of an electrical current.

### 4.6. Conditional Medium in Directing SC Fate in the Presence of the Carbon-Based Scaffold

SCs culture media (CM) plays a significant role in controlling SC behaviours by providing biochemical cues to SCs. CM provide the essential nutrients for the survival of cells and may serve as an instructor. A CM specific to cell type may also improve the scaffold’s response, which in turn may enhance cell adhesion, proliferation, and viability [153]. However, due to the diversity of SC types, a universal optimal SC CM has not been discovered yet, and more investigation is required to fulfil the needs or conditions of every type of SC [66]. The common ingredients of the medium used in SC culture are basal medium (e.g., Dulbecco’s modified eagle medium), serum (e.g., foetal bovine serum (FBS), platelet lysate), and antibiotic (e.g., penicillin/streptomycin), and usually cultured at 37 °C in 5% CO_2_ atmosphere [154,155]. Sometimes, antifungal drugs (e.g., amphotericin B and amino acids) or other additives (e.g., insulin) may be applied based on the purpose of the study [156]. Appropriate biological signals, such as the growth factors in a CM, are essential to understand the mechanisms regulating SC proliferation and differentiation. For example, osteogenesis stimulators, such as bone morphogenetic protein-2 (BMP-2) and transforming growth factor-β (TGF-β1) [157]. The addition of serum type for its function and effects on SC culture is still under debate. The use of different xenogeneic serums, such as FBS and foetal calf serum (FCS), is one concern that requires attention. Although the use of FBS and FCS in MSCs cultured in clinical studies does not show any significant secondary effect, there may be contamination from diseases and pathogens, which may give rise to immunological reactions [158,159]. Therefore, antibiotics and antifungals were added to the medium to prevent any possible microorganism contamination. Meanwhile, alternative factors to replace an animal-based serum medium of FBS or FCS with animal serum-free medium is under investigation.

Additionally, the interaction between the microstructure and CM is pivotal in building a suitable microstructure. Duan et al. reported the influence of different CM on the microstructure and nanomechanical properties of the mineralised matrix produced by the human MSC line Y201. In this study, the researchers observed that the basal medium promoted stiffer and more anisotropic microstructure of the mineralised matrix (i.e., bone nodule) compared to the osteogenic medium. The bone nodule in the basal medium also demonstrated a better nanomechanical response compared to the osteogenic medium. Unfortunately, both bone nodules from the basal and osteogenic media showed a reverse ageing effect in mechanical properties, which may be due to the higher cell proliferation rates than the mineralisation process [160]. Additionally, there is a concern involving the degradation of microstructure as a result of prolonged CM exposure. In an extended culture period, the scaffolds must be able to maintain their mechanical properties. Hence, the potential of carbon materials enables the production of cytocompatibility and patternability scaffolds with excellent mechanical strength, chemical inertness, and high swelling resistivity.

## 5. Application of Carbon-Based Scaffold in Tissue Engineering

The biocompatibility of carbon-based scaffolds with a variety of stem cells, including other biological cells, were positively significant. Numerous studies have also reported the potential of the carbon-based scaffolds in supporting and/or directing stem cells to differentiate into a variety of cell lineages.

### 5.1. Neural Tissue

Carbon-based scaffold in neural tissue engineering applications has shown significant potential in numerous studies. As such, Shin et al. has developed a scaffold incorporated with CNTs for neural tissue regeneration. They reported that the addition of CNTs improved the scaffold’s mechanical properties, swelling ability, and degradation rate. The scaffolds also enhanced the neuronal differentiation of human foetal neural stem cells (hfNSCs) and hiPSC-NPCs. However, an increase in CNT concentration led to significant cytotoxicity. Therefore, low CNT concentration is preferable to reduce the cytotoxicity effect and ensure cell viability [161].

Similarly, Hasanzadeh et al. developed a scaffold containing MWCNT for neural tissue engineering. The incorporation of MWCNT improved the electrical conductivity and mechanical properties of the scaffold in addition to, enhanced cell adhesion, proliferation, and the viability of human endometrial stem cells (hEnSCs) [162].

Meanwhile, Lee et al. developed a 3D printed scaffold incorporating amine-functionalized multi-walled carbon nanotubes (MWCNT) for neural tissue engineering. They reported that the addition of MWCNTs provided good electrical conductivity properties and improved the elastic modulus of the scaffold. The scaffold also supports cell adhesion and growth, as well as promotes the neuronal differentiation of NSCs. Furthermore, the electrical stimulation of cells enhanced the cells’ viability and neural differentiation by upregulating the neural marker of TUJ1 and GFAP [163].

In contrast, Chen et al. developed a Porphyra polysaccharide-based CD via one-pot hydrothermal treatment for non-viral gene carrier and neural induction. The resulting CDs were able to condense macromolecular plasmid DNA (CDs/pDNA). The CDs/pDNA nanoparticles enhanced the neural differentiation of ectodermal MSCs better than CDs alone. They also reported that the cellular uptake of CDs/pDNA occurred in multiple pathways, including clathrin and caveolae-dependent endocytosis. The multiple pathways of CDs/pDNA cellular uptake may improve the transfection efficiency of CDs/pDNA, thus enhancing the neural differentiation of ectodermal MSCs [61].

### 5.2. Cardiac Tissue

Other than neural tissue application, the carbon-based scaffold also significantly influenced cardiac tissue regeneration. For instance, Mombini et al. developed an electrically conductive chitosan-PVA-CNT nanofibers scaffold by using the electrospun technique for cardiac tissue engineering application. They reported that the addition of CNT improved the electrical conductivity, mechanical properties, and chemical stability of the resulting scaffold. It also improved the adhesion, growth, and viability of MSCs, as well as enhanced cardiac differentiation of MSCs on the scaffold [127].

Meanwhile, Yan et al. (2020) developed scaffolds containing P-phenylenediamine surface functionalised carbon quantum dots (CQDs) from graphite rods. The addition of CQDs improved the compressive modulus and swelling properties of the scaffold. It also enhanced the metabolic activity and viability of rat cardiomyocytes. Plus, it upregulated the cardiac-marker gene [164].

Moreover, Martinelli et al. developed a3D carbon nanotube composites scaffold by incorporating MWCNTs into the scaffold. They reported that the scaffold improved the neonatal rat ventricular cardiomyocytes (NRVM) viability, proliferation, and maturation to cardiac myocytes, while subduing the proliferation of cardiac fibroblast [165].

### 5.3. Bone Tissue

Interestingly, the carbon-based scaffold was able to support and promote bone tissue regeneration. For instance, Tohidlou et al. developed a scaffold incorporated with amine-functionalised single-walled carbon nanotube (aSWCNT) for bone tissue engineering. They reported that the addition of aSWCNT improved the scaffold’s tensile strength, electrical conductivity, bioactivity, and degradation rate. Furthermore, it also enhanced the attachment, proliferation, and differentiation of rat BMSCs [166].

In contrast, Nie et al. developed a 3D scaffold containing reduced graphene oxide (RGO) for bone tissue engineering. They reported that the scaffold with RGO improved the in vitro rat BMSCs adhesion, proliferation, and osteogenic differentiation. Additionally, in vivo study showed that the scaffold positively promoted the healing of circular calvarial defects in rabbits in 6 weeks with enhanced collagen deposition, cell proliferation, and mineralisation of new bone formation [167].

Meanwhile, Shafiei et al. (2019) developed a scaffold incorporated with carbon dots (CDs) by electrospun techniques. They reported that the synergetic effect of CDs and calcium phosphate on the scaffold enhanced the metabolic activity and proliferation of human buccal fat pad-derived stem cells (hBFPSCs). It also promoted a higher osteogenic differentiation and proliferation rate of hBFPSCs [168].

In addition, Amiryaghoubi et al. developed an injectable thermosensitive scaffold containing graphene oxide for bone tissue engineering. They reported that the addition of graphene oxide improved the scaffold’s mechanical properties, swelling abilities, and degradation rate. Additionally, the scaffold was haemocompatible and promoted the growth and viability of human dental pulp stem cells (hDPSCs). This also supports and enhances the osteogenic differentiation of hDPSCs [169].

Moreover, Dai et al. developed a 3D chitosan/honeycomb porous carbon/hydroxyapatite (CS/HPC/nHA) for bone tissue engineering (Figure 9). They showed that the addition of HPC improved the swelling abilities and mechanical properties of the scaffold. The resulting scaffold also promotes the growth, proliferation, viability, and osteogenic differentiation of mouse BMSCs. Meanwhile, the in vivo study of the scaffold has significantly promoted bone regeneration on distal femoral condyle defects of the rabbit model [170].

### 5.4. Others Tissue

The carbon-based scaffold’s ability to promote stem cell differentiation has motivated another researcher to instigate the ability of the carbon-based scaffold in another area of tissue engineering application. For instance, Tondnevis et al. has developed a scaffold incorporated with SWCNT for dental tissue engineering. They reported that the presence of SWCNT in the scaffold production by electrospun techniques caused bead formation. Also, the incorporation of SWCNT in the scaffold affects the drug’s release rate, allowing the prolonged and continuous release of the drug during regeneration. Moreover, it improved the hDPSC’s adhesion and proliferation on the scaffold [171].

Meanwhile, Gopinathan et al. developed a freeze-dried scaffold incorporated with carbon nanofiber (CNF) for meniscal tissue engineering. The addition of CNF improved the mechanical properties of the scaffold. In vitro study of the scaffold containing CNF also promotes cells adhesion, proliferation, and viability. Meanwhile, the scaffold biotoxicity study on rabbits showed that the scaffold was non-toxic [172]. However, Stocco et al. reported that their scaffold reinforced with CNT improved the mechanical properties, but it does not influence MSC survival [126].

Aspiringly, Yang et al. developed a scaffold incorporated with CNTs for retinal tissue regeneration. They reported that the addition of CNTs improved the scaffold degradation rate and electrical conductivity properties by 16.46%. Furthermore, the increased CNT concentrations, up to 50 µg/mL, do not affect the survivability of the cells. Moreover, in vitro study on the CNT scaffold showed enhanced cell adhesion and migration. It also supported BV2 cells and retinal ganglion cells (RGCs). Plus, the scaffold promoted hiPSCs differentiation into retinal ganglion cells [173].

## 6. Conclusions and Future Perspective

The interaction between SCs and their environment is quite well-evidenced; however, the extent to which this interaction controls the fate of SCs is still unclear. Moreover, the microenvironment niche and components in SC niches vary for each type of SC [174,175]. Plus, achieving a similar effect of cells in vivo remains a challenge in SCs studies. Furthermore, it is crucial to develop culture conditions that will promote the homogenous and enhanced differentiation ability of SCs into functional and desired tissues. Therefore, understanding the SCs characteristics and regulatory mechanisms is vital in creating a suitable microenvironment niche for SCs. This approach is also essential for SC efficacy and safety in clinical application. Furthermore, the scaffold design plays the most critical role in SC applications, as it regulates the behaviour of SCs, leading to cell lineage differentiation.

However, the development of scaffolds, which mimic the ECM of the SC natural microenvironment niche, is not a simple task. Many parameters should be accounted for when producing a suitable scaffold. These include biophysical and biochemical signals, the extracellular microenvironment, and proper guidance of SC behaviours which are crucial factors that require consideration to achieve a thriving SC culture on the carbon-based scaffold. Thus, patternable and suitable biomaterials should be selected accordingly. The carbon material provides patternable materials (i.e., carbon precursors such as a polymer). Also, carbon materials possess excellent chemical inertness, electroconductivity, mechanical strength, and swelling resistivity. However, carbon still has its limitations that can affect SC response. Thus, further investigation may enhance carbon-based scaffolds as biomaterials for SC application.

Regarding this, technological advancement, such as additive manufacturing methods enabling the fabrication of a complex scaffolds with precise structural designs, is required. Additive manufacturing allows a 3D printing application on a wide range of biomaterials. This method may also allow the fabrication of a scaffold similar to the ECM of SCs, and other biological cells, as it provides a controllable structure production. Currently, various biomaterials, including natural and synthetic polymers, have been investigated as ink for 3D printing and the resulting products are biocompatible to a wide range of cells. Unfortunately, even though manufacturing scaffold with additive manufacturing method is advantageous, the cost of the device and setup remain a challenge. Plus, modification is needed on the natural polymer for it to be suitable as ink for 3D printing. Therefore, additive manufacturing may not be easily accessible due to the cost of the device and materials required to investigate the optimal ink for them to be 3D printable. However, the growing interest in additive manufacturing is increasing due to its potential for modulating the SCs microenvironment and enhancing its development. Hence, the existence of additive manufacturing allows the possibility to develop the most optimal microenvironment specifics to SCs.

## Figures and Tables

**Figure 1 polymers-13-04058-f001:**
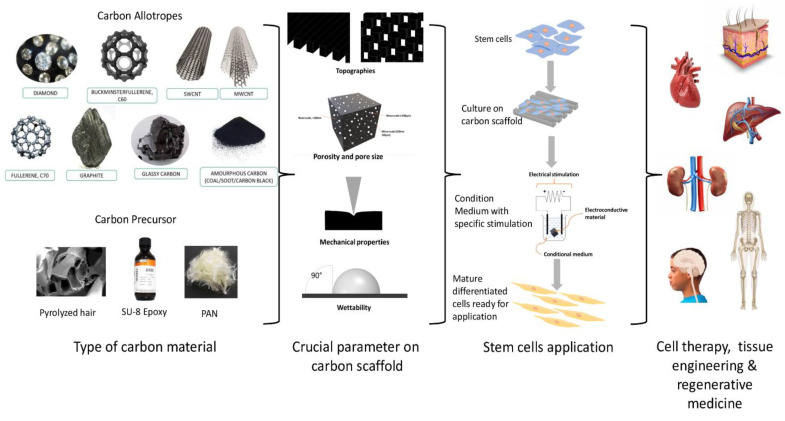
Schematic illustration of carbon material application.

**Figure 2 polymers-13-04058-f002:**
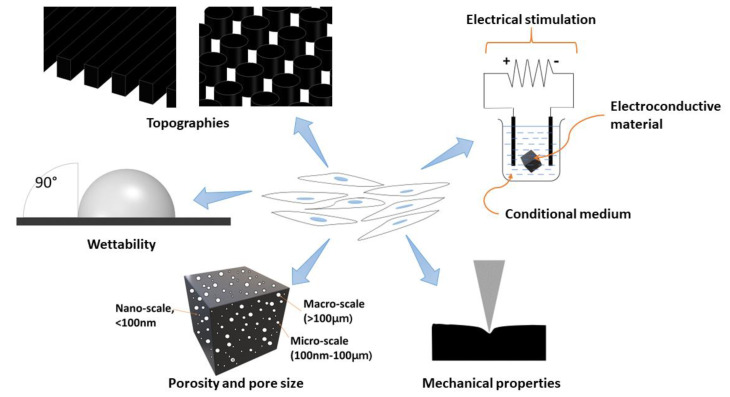
Schematic illustration of factors influencing the behaviours of stem cells.

**Figure 3 polymers-13-04058-f003:**
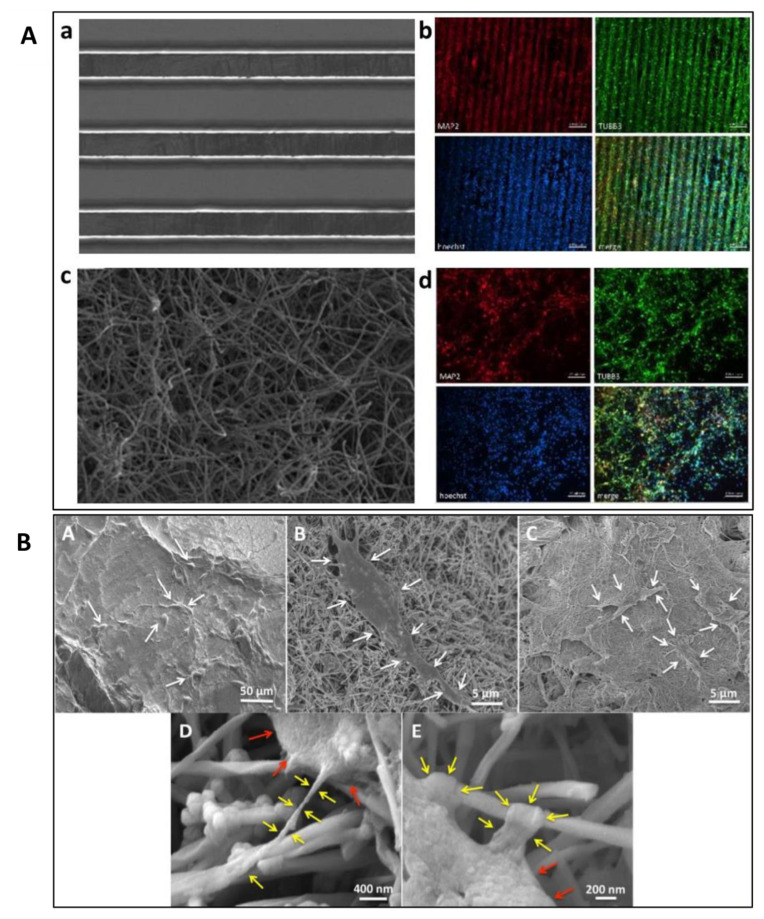
(**A**) SEM and fluorescent microscopy images of human NSC-derived neurons cells on pyrolyzed aligned SU-8 (a,b) and PAN fibres (c,d). Reproduced with permission from Ref. [90]; copyright (2020), Elsevier. (**B**) SEM images of ADSC adhesion on (A) PLGA, (B) MWCNT, and (C) SWCNT scaffolds. Magnification images of ADSCs on MWCNT scaffolds show cytoplasmic extensions (yellow arrows) attached to the nanotube (D,E). Red arrows indicate cell body. Reproduced with permission from Ref. [98]; copyright (2017), John Wiley and Sons.

**Figure 4 polymers-13-04058-f004:**
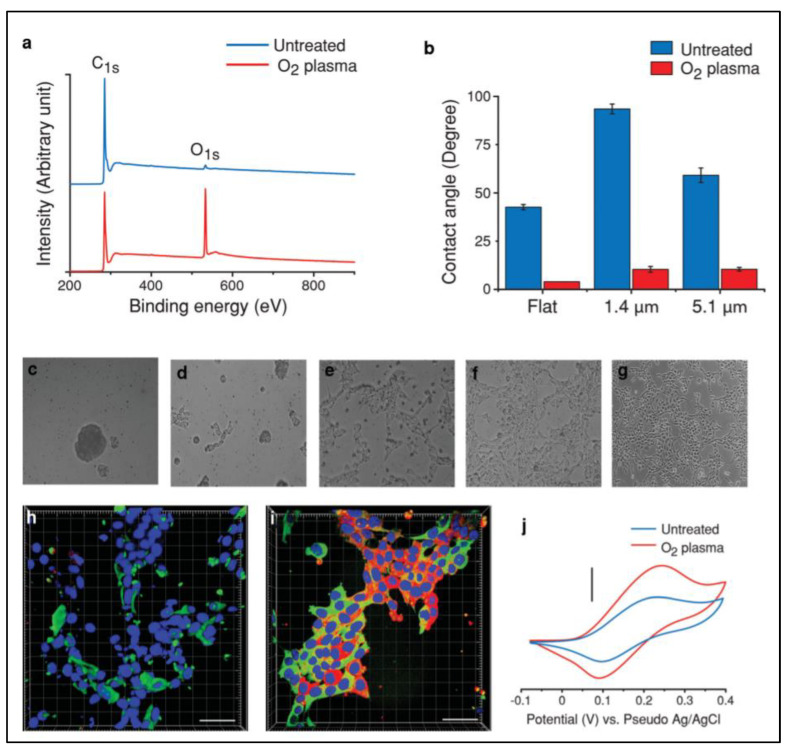
XPS characteristics of flat carbon before and after oxygen plasma treatment (**a**). Contact angle measurements on flat carbon and p3D-carbon (**b**). (**c**–**g**) Representative bright field images of hNSCs cultured in the presence of growth factors (48 h) on (**c**) untreated, (**d**) oxygen plasma-treated, (**e**) PLL-coated, (**f**) oxygen plasma-treated and PLL-coated flat carbon surfaces, and (**g**) PLL-coated TCPS. (**h**,**i**) Representative confocal fluorescence images of human NSCs cultured in the presence of GF (48 h) on (**h**) PLL-coated TCPS and (**i**) oxygen plasma-treated and PLL-coated flat carbon surfaces. Characteristic cyclic voltammograms of dopamine (5 mM) in PBS (pH 7) on a p3D-carbon before and after oxygen plasma treatment (**j**). Reproduced with permission from Ref. [81]; copyright (2014), John Wiley and Sons.

**Figure 5 polymers-13-04058-f005:**
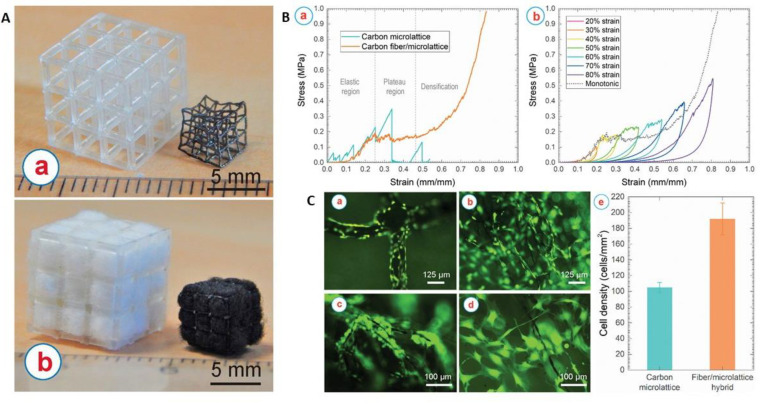
(**A**) (a) Additively manufactured microlattice architectures before (left) and after (right) carbonisation. (b) Cotton/microlattice hybrid architectures before (left) and after (right) carbonisation. (**B**) (a) Stress–strain curve of the carbon microlattice, and the carbon fiber/microlattice hybrid architecture. (b) Cycling loading of the carbon fiber/microlattice hybrid architecture with a sequential increment of strain. (**C**) Osteoblast-like murine MC3T3-E1 cells cultured on (a) carbon microlattice architectures and within (b–d) carbon fiber/microlattice hybrid architecture. (c) Proliferation of the cells along the carbon fibers and (d) shows the inter cellular network within the inner pores created by the carbon fibers. Some cells appear out of focus due to the presence of cells growing on different planes of the 3D scaffolds. (e) Comparison of the density of the cells colonised on the carbon microlattice architecture and within the carbon fiber/microlattice hybrid architecture. Reproduced with permission from Ref. [125]; copyright (2021), Elsevier.

**Figure 6 polymers-13-04058-f006:**
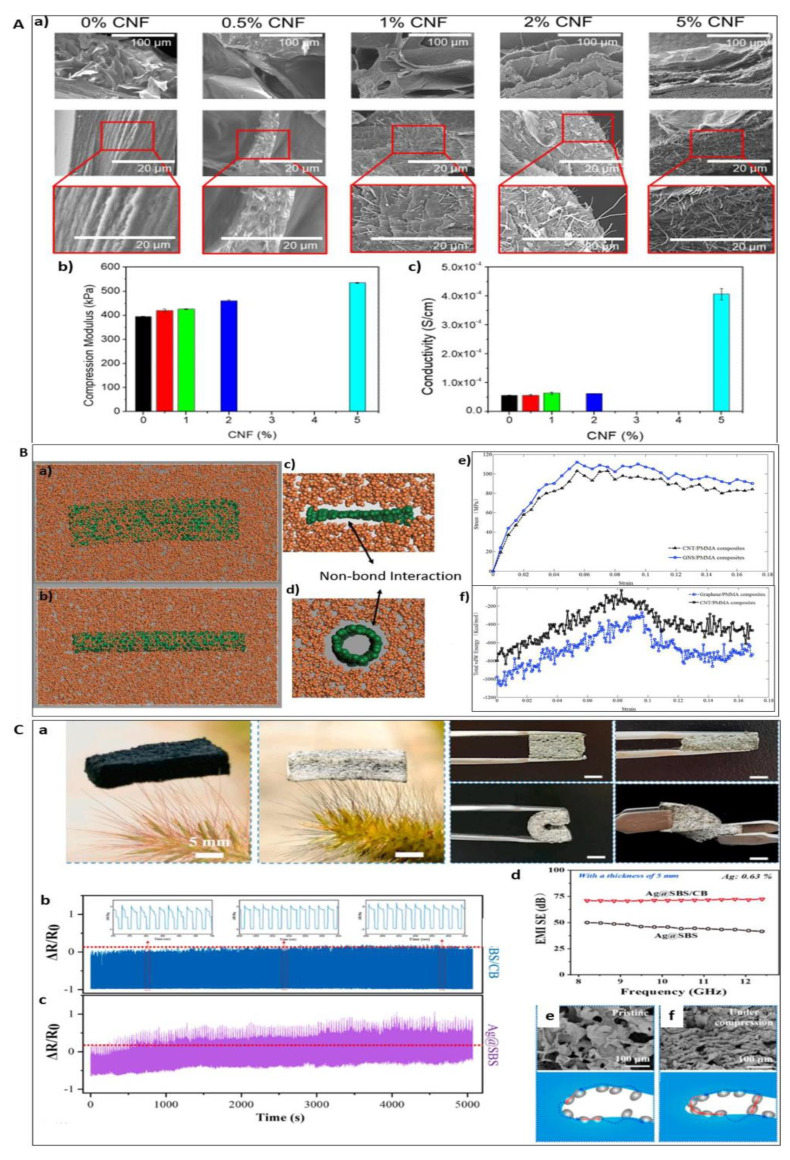
(**A**) SEM images of lyophilised hydrogels with different CNF content (a) (Top row scale bar 100 μm, middle and bottom row 20 μm); (b) Young’s modulus and (c) electrical conductivity of the samples of alginate/gelatin/CNFs hydrogels as a function of the CNFs content, respectively. Reproduced with permission from Ref. [124]; copyright (2021), Elsevier. (**B**) Molecular models of the: (a) CNT/PMMA composites, (b) GNS/PMMA composites, (c) and (d) cross section views of the interfacial interactions between the nano-reinforcements and PMMA matrices. The PMMA matrices and nano-reinforcements are presented by the colors of orange and green, respectively. (e) The strain–stress curves of the CNT, GNS/PMMA composites. (f) Total vdW energy of the CNTs, GNS/PMMA composites during the tensile processes. Reproduced with permission from Ref. [130]; copyright (2018), Elsevier. (**C**) (a) Optical image of SBS/CB frame, Ag@SBS/CB hybrid foam, and optical images of the Ag@SBS/CB hybrid foam under compression, bending and twisting. The scalar bar is 5 mm. Cyclic compression-release test of Ag@SBS/CB (b) and Ag@SBS foam (c). (d) EMI SE comparison of Ag@SBS/CB and Ag@SBS foam. (e,f) SEM images and corresponding schematics of Ag@SBS/CB under pristine and compression state. Reproduced with permission from Ref. [131]; copyright (2021), Elsevier.

**Figure 7 polymers-13-04058-f007:**
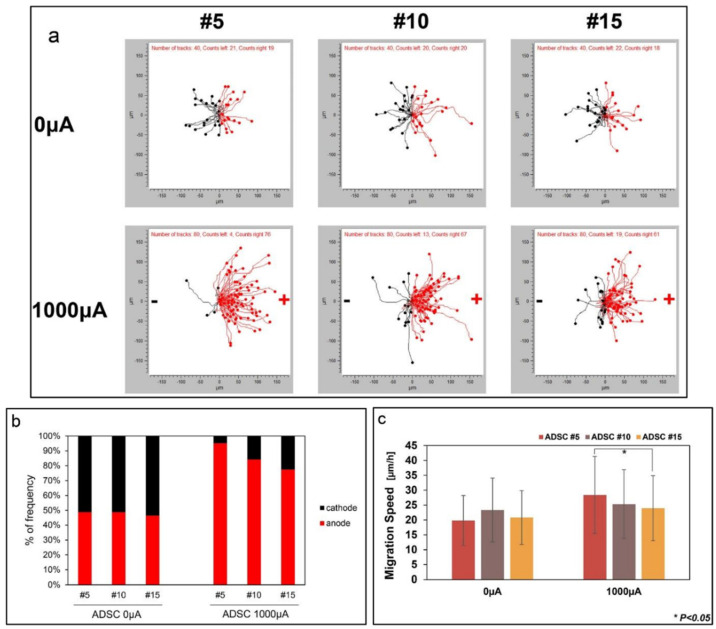
ADSCs migration by electrotaxis with increasing passage numbers. Reproduced with permission from Ref. [99]; copyright (2019), Elsevier. (**a**) ADSCs were manually tracked and cells migration was divided into the right (anode) and left (cathode). (**b**) Frequency of cell movement towards the anode and cathode. (**c**) Cell migration speed was measured at passages 5, 10, and 15 when direct current was applied to the cells. * *p* < 0.05.

**Figure 8 polymers-13-04058-f008:**
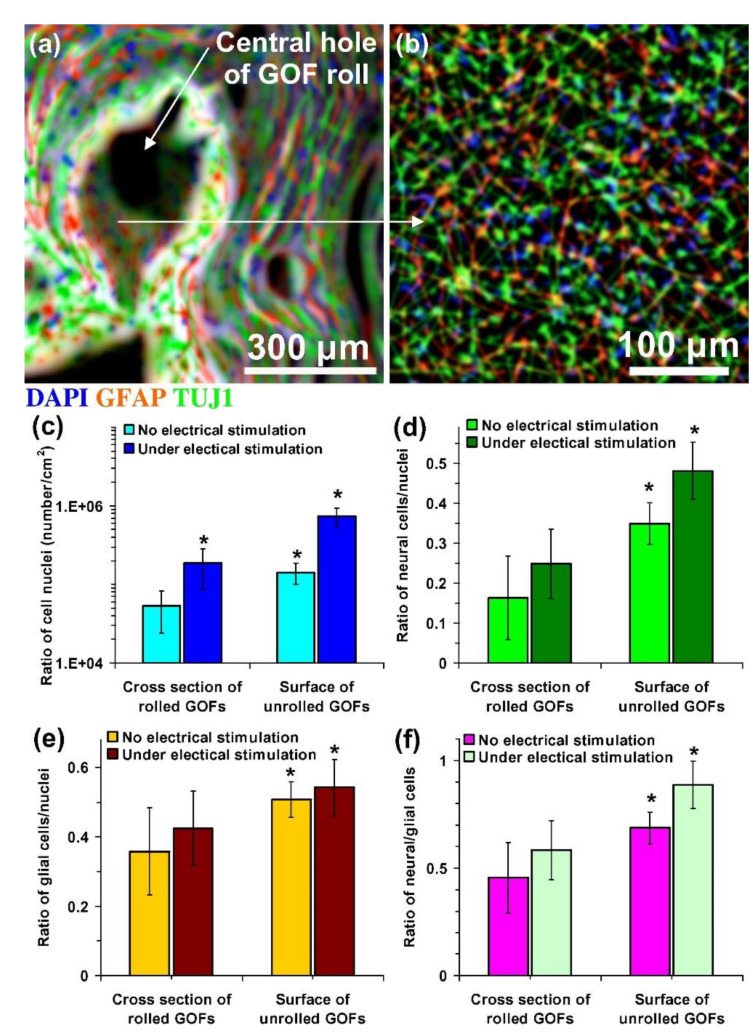
Fluorescent images of hNSCs differentiated on (**a**) cross-section and (**b**) interior surface of a rolled-GOF scaffold after two weeks of electrical stimulation. The nuclei, glial, and neural cells of the differentiated cells were stained by DAPI (blue colour), GFAP (red colour), and TUJ1 (green colour), respectively. (**c**) The number of cell nuclei per surface area of the images and ratio of the number of (**d**) glial cells (GFAP-positive cells) and (**e**) neural cells (TUJ1-positive cells) to the number of nuclei and (**f**) neural/glial cell ratios on cross-section and interior surface of the scaffolds after two weeks differentiation in the absence and the presence of electrical stimulation. Significant results are indicated by asterisks (*) for *p*-values < 0.05 (*n* = 5). Reproduced with permission from Ref. [138]; copyright (2016), Elsevier.

**Figure 9 polymers-13-04058-f009:**
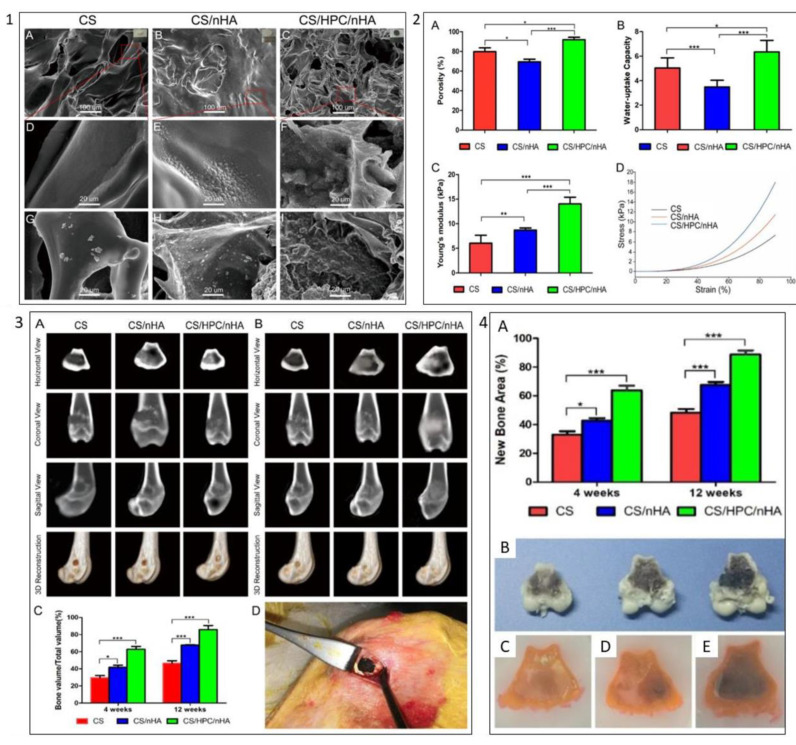
Example of carbon biomaterials effect on scaffold characteristics and BMSCs behaviours. (**1**) SEM images of (A,D) CS scaffold, (B,E) CS/nHA scaffold, and (C,F) CS/HPC/nHA scaffold, respectively; SEM images of (G) CS scaffold, (H) CS/nHA scaffold, (I) CS/HPC/nHA scaffold after 4 days immersion in simulated body fluid. (**2**) Scaffold characterisation: (A) Porosity, (B) uptake-water capacity, (C) elastic modulus, (D) typical stress–strain curves of the different scaffolds. (**3**) (A) CT images of the cross-section, coronal, sagittal, and three-dimensional reconstruction of the distal femoral defect area of the rabbit femur after 4 weeks of implantation of three different scaffolds. (B) 12 weeks postoperatively. (C) Morphometric analysis of the percentage of newly formed bone mass (BV/TV) at 4 and 12 weeks postoperatively. (D) Intraoperative photograph of the distal femoral condyle of the rabbit. (**4**) (A) Percentage of new bone area in the defects; (B) the cross-sectional morphology of the distal femoral defect area of the femur; tissue brace of the distal femoral defect area of the rabbit femur implanted with (C) CS, (D) CS/nHA and (E) CS/HPC/nHA scaffolds, (* *p* < 0.05, ** *p* < 0.03, *** *p* < 0.01). Reproduced with permission from Ref. [170]; Copyright 2020, America Chemical Society.

**Table 1 polymers-13-04058-t001:** Carbon materials application in SCs and other biological cells research.

Types of Carbon	Dimensions	Composite Material	Fabrication Methods	Types of Cells	Ref.
Carbon Nanocage	3D nanoscale	-	-	HUC-MSCs	[22]
Fullerene	Aligned fullerene nanowhisker nanopatterned	-	Langmuir–Blodgett	Human MSCs	[39]
	Aligned fullerene nanowhiskers	-	Modified liquid–liquid interfacial precipitation method	NSCs	[40]
Graphene	rGONRs grids	polydimethylsiloxane (PDMS)	Drop casting method	Human MSCs	[31]
	2D graphene (GNOs, GONRs, GONPs)	distearoyl-sn-glycero-3-phosphoethanolamine-N-[amino (polyethylene glycol)] (DSPE-PEG)	GONRs synthesis by using modified longitudinal unzipping method; GONPs synthesis by using modified Hummer’s method	AMSCs, BMSCs	[30]
	3D matrix	Polycaprolactone (PCL)	Extrusion-based additive manufacturing	AMSCs	[33]
	3D foams and 2D films	-	chemical vapor deposition (CVD)	NSCs	[41]
	Fibres	Poly-L-lactic-acid (PLLA)	Thermal-induced phase separation	BMSCs	[42]
	Nanosheets	PCL	Water-assisted liquid phase exfoliation	AMSCs	[43]
	3D graphene oxide	Polypeptide thermogel	Temperature-sensitive sol to gel transition	Tonsil-derived MSCs	[44]
	3D graphene	Nickel foam	CVD	Mouse NSCs	[45]
	3D Graphene/SWCNT	-	CVD	Mouse MSCs	[46]
Carbon Nanotube	COOH-SWCNT and -MWCNT, PEG-SWCNT	Ethanol, polyethylene Glycol (PEG)	Air brush spraying on a coverslip	Canine MSCs	[20]
	CNT fibres	PLLA	Thermal-induced phase separation	BMSCs	[42]
	CNT	PCL	CVD	AMSCs	[43]
	MWCNT	PCL	Electrospinning	Human Dental Pulp Stem Cell	[47]
	MWCNT	Thermoplastic polyurethane	Electrospinning	Rat AMSCs	[48]
	MWCNT	PLLA	Electrospinning	Mouse ESCs	[49]
	MWCNT	Collagen hydrogel	Gelation	Rat MSCs	[50]
	MWCNT	Polyion complex hydrogel	Extrusion-based 3D printing	Rat BMSCs	[51]
	MWCNT	PEG	Drop-drying method	Human MSCs	[52]
	MWCNT	Poly-lactic acid (PLA), alginate, gelatine	Layer-by-layer assembly method	Wharton’s Jelly-derived mesenchymal stem cells (WJMSCs)	[53]
Nanodiamond	Monolayer	-	Ultrasonication	Human NSCs	[54]
Reticulated vitreous carbon	3D Foam	-	Etching and Pyrolysis	BMSCs	[55]
Carbon Nano-onions	Poly 4-mercaptophenyl methacrylate-carbon nano-onions	PCL	Probe sonication, hydraulic pressing	Human osteoblast cells	[28]
	Oxidized CNOs	Chitosan, poly(vinyl-alcohol)	Cure on acetate molds	In vivo study on Wistar rat	[56]
	Poly 4-mercaptophenyl methacrylate-carbon nano-onions	Bovine serum albumin, trifluoroacetic acid	Force spinning	Human fibroblast cells	[57]
	Poly 4-mercaptophenyl methacrylate-carbon nano-onions	Gelatin	Probe sonication, freeze drying	Human osteoblast cells	[58]
Carbon black nanoparticle	Nanoparticles	-	Probe sonication	In vivo study on mouse brains astrocyte	[59]
Carbon dots	Citric acid-derived nanodots	-	Hydrothermal	Rat BMSCs	[60]
	Porphyra polysaccharide-derived carbon dots	-	Hydrothermal	Ectodermal MSCs	[61]
	Cellulose-derived reduced nanographene oxide carbon nanodots	PCL	Microwave	MG63	[62]
	Onion-derived carbon nanodots	-	Microwave	Human foreskin fibroblast, MG63, red blood cells	[63]
	Human fingernail-derived carbon nanodots	-	Pyrolysis	HEK-293	[64]
	Food-derived carbon nanodots	Glass beads	Hydrothermal	Prostate cancer (PC3) cells, NRK cells	[65]

**Table 2 polymers-13-04058-t002:** Carbon precursor in biological applications.

Type of Precursor	Fabrication Method	Structure	Application	Ref.
Citric acid	Hydrothermal	Carbon nanodots	Rat BMSCs	[60]
Porphyra polysaccharide	Hydrothermal	Carbon nanodots	Ectodermal MSCs	[61]
Polyacrylonitrile (PAN)	Electrospun, pyrolysis	Electrospun carbon nanofibres	Mouse NSCs culture	[23]
	Electrospun, pyrolysis	Electrospun carbon nanofibres	Human endometrial stem cells (hEnSCs)	[82]
Cryogel (chitosan/agarose/gelatin)	Pyrolysis	3D carbon-based scaffold	NSCs	[70]
Sucrose	Sugar blowing technique, Pyrolysis	3D glassy carbon	SH-SY5Y, HEK-293	[72]
Polydopamine	Electrospun, pyrolysis	Microfibre scaffold	NSCs	[83]
SU-8	Photolithography, pyrolysis	3D carbon-based scaffold	Human NSCs, PC12	[81]
	photolithography, Pyrolysis	Gold nanoparticles glassy carbon	Primary dermal fibroblast	[84]
Zif-8	Pyrolysis	C-ZnO nanoparticles	MSCs	[85]
Cotton	Pyrolysis	Pyrolysed cotton microfibres	PC12	[86]
Epoxy resin	Stereolithography, pyrolysis	Carbon microlattices	MC3E3-E1	[87]

**Table 3 polymers-13-04058-t003:** Topographies influence the behaviour of SCs on a scaffold.

Materials	Fabrication Methods	Topography	Type of Cells	Outcomes	Ref.
Reticulated vitreous carbon	Etching and pyrolysis	Foams with tantalum coating	BMSCs	The scaffold promotes adhesion, aggregation, and proliferation of BMSCs.	[55]
Cryogel-derived carbon	Pyrolysis	3D glassy carbon patterned	Human NSCs	Cells cluster reside within the porous structure.	[70]
SU-8-derived carbon	Photolithography and pyrolysis	Pillar patterned	Human NSCs	On the pillar, cells showed the elongation neurites formation, which does not show on the flat carbon.	[81]
PAN-derived carbon nanofibers (CNFs)	Electrospinning and pyrolysis	Aligned patterned and random alignment	hEnSCs	Lower cell proliferation on aligned CNFs compared to random CNFs. Plus, upregulation of cardiac marker, NF-H, and Tuj-1, and downregulation of nestin on aligned CNFs compared to random CNFs. Moreover, on aligned CNFs, the differentiated cells extended along the CNF central axis, whereas on random CNFs, the cells stretched multi-directionally.	[82]
SU-8 and PAN	Photolithography of SU-8, electrospinning PAN, pyrolysis	Aligned patterned and random alignment	Human induced pluripotent stem cells derived neural stem cells (hiPSCs-NSCs)	Cells on aligned SU-8 show higher gabaergic and cholinergic neuron differentiation compared to on PAN fibres. Also, on SU-8, cells aligned along lines sidewalls surface, forming long cytoskeleton protrusions, whereas on PAN fibres, cells spread randomly on the available surface with the random spreading of cytoskeleton protrusions.	[90,94]
SU-8	Photolithography and pyrolysis	Aligned patterned	hiPSCs-NSCs	The aligned pattern strongly influenced the arrangement and orientation of the cells.	[95]
PCL	Electrospinning	Random alignment (REF), mesh-like alignment (MEF), and align patterned (AEF)	AMSCs, human umbilical vein endothelial cells (HUVECs)	On REF, the cells were round with protruding edges. Whereas on AEF and MEF, the cultured cells were elongated and oriented onto the aligned or bundled fibres. On MEF, the cells arranged themselves both on the grid lines on the fibre bundles and inside the grids on the loose fibres.	[96]
SU-8, tetrahedral amorphous carbon (ta-C)	Etching, thin film deposition, photolithography, pyrolysis	Planar, nanopillars, micro-pillars, and nano-rough patterned	Mouse NSCs	On SU-8, no significant changes of cell viability were observed on every pattern. Whereas On ta-C, higher cell viability on 2 nm pillars compared to planar and 20 µm pillars. Meanwhile, on 2 µm pillar height, higher cell count, and aggregation of cells on the ta-C scaffold were observed compared to SU-8.	[97]

**Table 4 polymers-13-04058-t004:** Surface modifications on carbon-based scaffold.

Surface Modification	Types of Cells	Substrates	Methods	Ref.
**Mineral deposition**	BMSC	Reticulated vitreous carbon	Tantalum coating	[55]
**Surface functionalisation**	AMSCs	PCL/graphene	NaOH treatment	[33]
	Mouse ESCs	PLLA-MWCNT and PLLA-SWCNT nanofiber scaffold	Plasma treatment	[49]
	Human MSCs	PEG/MWCNT	MWCNT treated with HNO_3_ and H_2_SO_4_ (1:3 *v*/*v*)	[52]
	Human NSCs	Nanodiamond	Hydrogen and oxygen treatment	[54]
	Menstrual derived-SCs	PLA/MWCNT	MWCNT treated with HCL and HNO_3_	[80]
	Human NSC	SU-8-derived carbon	Oxygen plasma treatment	[81]
**Topographical modification**	Human NSC	SU-8-derived carbon	Pillar patterned	[81]
	hiPSCs -NSCs	SU-8-derived carbon	Align patterned	[90]
	Human MSC	Coverslip	Spray pegylated MWCNT and covalently immobilization of bone morphogenetic-2 (BMP-1)	[104]
**Protein absorption**	C6 rat glial cells, PC12, rat neuroblastic cells	SU-8 derived carbon	Poly-l-lysine	[105]
	Human ostoblast cells	PCL/CNOs	Poly 4-mercaptophenyl methacrylate	[106]

**Table 5 polymers-13-04058-t005:** The effect of pore size on SCs behaviours.

Pore Diameter of Study	Porosity	Cells of Study	Outcomes	Ref.
≈90 µm to 400 µm.	Approx. 80–90%	ADSCs	The pore size range of 370–400 µm was more favourable by ASCs for chondrogenic differentiation than the other pore size group.	[110]
100, 200, and 400 µm.	-	ADSCs	After 21 days, the cells showed proliferation and migration on the scaffold with 100 µm and 200 µm pore size, whereas lumps of cells were present on the scaffold with a 400 µm pore size. The scaffold with the 200 µm pore size showed better cell proliferation and cell–scaffold interaction than the scaffold with 100 µm or 200. Whereas scaffold with the 400 µm pore size showed the most significant chondrogenic differentiation.	[111]
7, 12, and 17 µm.	-	MSCs	Scaffolds with a pore size of 12 µm showed higher MSC migration rates than other pore sizes.	[112]
125–300 µm, 300–500 µm, and 500–850 µm.	-	Human MSCs	The scaffold with a pore size of 500 to 850 µm stimulated the highest osteogenic response.	[113]
≈830 µm of cubic pore and ≈730 µm of the cylindrical pore.	≈80%	Human MSCs	The cubic pore geometry promotes osteogenesis. Whereas the cylindrical pore geometry promotes adipogenic and chondrogenic.	[114]
≈94 µm, ≈130 µm, and ≈300 µm.	-	Rat BMSCs	Increasing the pore size led to increasing cell attachment ability. Higher cell density in 300 µm pore group compared to 94 µm. Flat cell morphology appears on 94 µm and 130 µm pore group, while the 300 µm pore group cells appear in rounded morphology. The largest pore size of 300 µm stimulated the highest cell proliferation, chondrogenic gene expression, and cartilage-like matrix compared to the smaller pore sizes of 94 µm and 130 µm.	[115]
≈173.8 µm, ≈275.23 µm, ≈384.52 µm.	≈83.87%, ≈87.03%, ≈95.28%	Porcine BMSCs	Smaller pore size scaffold (173.8 µm) shows the highest osteogenic differentiation of BMSCs compared to enormous pore sizes scaffold.	[116]
≈4060 µm, ≈6330 µm, and ≈7600 µm.	-	Neural progenitor stem cells (NPSCs)	The cell differentiation rate increased alongside the increased pore size scaffolds. Whereas the total cell numbers decreased with increasing porosity.	[117]
150 µm, 200 µm, and 250 µm	54%, 60%, 65%	Human BMSCs, human ADSCs	Higher matrix mineralization on 150 µm pore size scaffold compared to another pore size scaffold. Whereas the lowest cell numbers were presented on a 200 µm pore size scaffold.	[118]

**Table 6 polymers-13-04058-t006:** Electrical stimulation for SCs application.

Type of Cells	Methods	Outcomes	Ref.
BMSCs	Direct current electrical fields of 200 mV/mm and 600 mV/mm for 2–10 h at 37 °C	Without ES, the cells migrated randomly, whereas the cells move towards the anode with ES.	[134]
Human AMSCs	1.7 V AC/20 Hz for 24 h, 72 h, 7 days	At 72 h, ES enhanced the growth and proliferation of cells. However, at seven days, the enhancement was reversed.	[140]
Human NPSCs	ES with + 1 V to −1 V square wave at 1 kHz for 1 h	ES of human NPSCs led to the changes of the VEGF-A pathway and genes involved in cell survival, inflammatory response, and synaptic remodelling.	[141]
BMSCs, AMSCs	100 mV/mm of DC ES for 1 h per day	In BMSCs, ES increased mRNA levels of Runx 2, osteopontin, and Col1A2 at day 7. Whereas in AMSCs, ES increased Runx2 and osteopontin expression observed after 14 days.	[142]
hiPSCs	65 mV/mm and 200 mV/mm at 1 Hz frequency and 1 ms pulse width for 1.5, 5, 10, and 15 min durations	On day 2, spontaneously beating hiPSCs was obtained. At acute 5 min ES, higher beating embryoid bodies at day 14 were obtained compared to the other duration of the study (1.5, 10, and 15 min). Also, ES could stimulate hiPSCs cardiac differentiation. The cardiogenic effect of acute ES was cell line dependent.	[143]
hiPSCs	1 V/cm and 1.5 V/cm with a biphasic pulse (5 ms) at 5 Hz frequency for 1–30 days	Under ES, spontaneously beating of hiPSCs were observed as early as two days. Also, ES enhanced the cardiac differentiation of hiPSCs and promoted cardiomyocyte maturation.	[144]
Human AMSCs, HUVECs	200 µA for 4 h/day	ES promote osteogenic differentiation.	[145]
Rat MSCs	Triboelectric nanogenerator (TENG) electrical signal at 3000 pulses/day with an output of 300 V and 30 µA with a frequency of about 120 times/min.	Improved neural differentiation of MSCs.	[146]

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
