# Peer review of "Modelling of Stem Cells Microenvironment Using Carbon-Based Scaffold for Tissue Engineering Application—A Review"

_polymers, 2021, doi:10.3390/polym13234058_

Round 1

Reviewer 1 Report

This review presents a great study of carbon scaffolds used widely in the literature. Thus, this work is novel and could be interesting for the readers of this journal. Nevertheless, it is necessary a modification before its publication: Mechanical section must be improved with more specific data of scaffolds obtained in literature.

Author Response

Thank you for the comments. The manuscript has been revised based on the comments given. (page 15, lines 384-416).

Reviewer 2 Report

In this manuscript (polymer-1460452), the authors have reviewed the efficacy of carbon-based scaffolds for modulating stem cells microenvironment in tissue engineering applications. This review manuscript is well written and presented using comprehensive literature. While gone throughout the manuscript, I have found that this manuscript can be accepted for publication after a minor revision.

  1. Authors are suggested to cite some more recently published articles in Introduction section on carbon-based functional scaffolds for tissue engineering applications. For example, Composites Science and Technology, 175, 2019, 35-45; Carbohydrate polymers, 193, 2018, 228-238; Frontiers in Materials, 7, 2020, 1-10; International Journal of Biological Macromolecules, 180, 2021, 590-598; ACS Biomaterials Science & Engineering7, 2021, 2627-2637; etc.
  2. In conclusion and future perspective section, please discuss more on the effect of additive manufacturing methods on modulating SCs microenvironment with challenges and opportunities.

Reviewer 3 Report

Please see the attachment for the reviewer's comments. 

Round 2

Reviewer 3 Report

The authors have clarified all my concerns, and the quality of the manuscript was significantly improved. I must congratulate the authors for their willingness in addressing the reviewer’s comments. So, I recommend accepting the manuscript in its present form.